# Validation of Copernicus Sentinel-3/OLCI Level 2 LAND Integrated Water Vapour product

Niilo Kalakoski[1], Viktoria F. Sofieva[1], René Preusker[2], Claire Henocq[3], Matthieu Denisselle[3], Steffen Dransfeld[4], and Silvia Scifoni[5]

[1]Finnish Meteorological Institute, Helsinki, Finland
[2]Institute for Space Sciences, Freie Universität Berlin (FUB), Germany
[3]ACRI-ST, Sophia-Antipolis, France
[4]European Space Research Institute (ESRIN), Frascati, Italy
[5]Serco Italia SpA for European Space Agency (ESA), European Space Research Institute (ESRIN), Frascati, Italy

**Correspondence:** Niilo Kalakoski (niilo.kalakoski@fmi.fi)

**Abstract.**

Validation of the Integrated Water Vapour (IWV) from Sentinel-3 Ocean and Land Colour Instrument (OLCI) was performed as a part of "ESA/Copernicus Space Component Validation for Land Surface Temperature, Aerosol Optical Depth and Water Vapour Sentinel-3 Products" (LAW) project. High spatial resolution IWV observations in the near infrared spectral region from the OLCI instruments aboard Sentinel-3A and -3B satellites provide continuity with observations from MERIS (Medium Resolution Imaging Spectrometer). The IWV was compared with reference observations from two networks: GNSS (Global Navigation Satellite System) derived precipitable water vapour from the SUOMINET network and integrated lower tropospheric columns from radio-soundings from the IGRA (Integrated Radiosonde Archive) database.

Results for cloud-free matchups over land show a wet bias of 7-10% for OLCI, with a high correlation against the reference observations (0.98 against SUOMINET and 0.90 againts IGRA). Both OLCI-A and -B instruments show almost identical results, apart from an anomaly observed in camera 3 of the OLCI-B instrument, where observed biases are lower than in other cameras in either instrument. The wavelength drift in sensors was investigated, and biases in different cameras were found to be independent of wavelength. Effect of cloud proximity was found to have almost no effect on observed biases, indicating that cloud-flagging in the OLCI IWV product is sufficiently reliable. We performed validation of random uncertainty estimates and found them to be consistent with the statistical a posteriori estimates, but somewhat higher.

## 1 Introduction

Total column water vapour (TCWV) is one of the essential climate variables defined by the GCOS (Global Climate Observing System) Climate Monitoring Principles (Bojinski et al., 2014). On large temporal and spatial scales, water vapour is a strong greenhouse gas, contributing to radiative climate feedback loops. Water vapour contributes also to climate and weather processes through latent heat transport (eg. Bengtsson, 2010). On smaller spatio-temporal scales, amount of water vapour in the atmosphere affects local weather conditions and hydrological cycles (Bengtsson and Hodges, 2005; Sherwood et al., 2010).

Because of the importance of water vapour for the climate and weather predictions, TCWV, also referred as Integrated Water Vapour (IWV) or Total Precipitable Water (TPW), has been continuously observed for decades by wide range of methods. In addition to ground-based and in-situ observations, satellite observations using passive imagers on polar-orbiting satellites can provide daily near-global coverage. The longest time-series are available from measurements in microwave region, with SSM/I and SSMIS instruments on various satellite platforms. Advantage of the microwave observations is the ability to provide the TCWV in both cloud-free and cloudy conditions. However, microwave observations are generally restricted to water surfaces (Schluessel and Emery, 1990; Wentz, 1997). In thermal infrared spectral region, water vapour can be retrieved from broadband radiometers (e.g., HIRS (Shi and Bates, 2011)) or from spectrometers (e.g., AIRS, IASI (Schlüssel and Goldberg, 2002; Roman et al., 2016)). Observations by broadband radiometers provide another source of long timeseries (since early 1980's). Hyperspectral infrared observations with improved vertical information and reduced uncertainty are available from early 2000's. Retrievals of TCWV from infrared observations alone are only possible in clear-sky, or nearly clear-sky conditions. (Schröder et al., 2018). Spectrometers operating in UV and visible spectral region can yield TCWV observations in clear-sky conditions over land and ocean. Modified DOAS (Differential optical absorption spectroscopy) has been applied to GOME (Noël et al., 1999), SCIAMACHY (Noël et al., 2004), GOME-2 (Noël et al., 2008; Grossi et al., 2015) and Sentinel-5P (Küchler et al., 2022) observations. Wagner et al. (2013) and Wang et al. (2014) used GOME-2 and OMI observations, respectively, to retrieve TCWV in blue spectral range, taking advantage of higher and more homogeneous surface albedo in that range.

In the near infrared, water absorption lines wavelengths around 900 nm can be used retrieve TCWV. Similarly to UV/VIS instruments, retrievals from NIR instruments have advantage of relatively low dependency on surface type over land. In addition, retrievals from MERIS (Medium Resolution Imaging Spectrometer, e.g. Lindstrot et al. (2012)) and MODIS (Moderate Resolution Imaging Spectroradiometer, e.g. Diedrich et al. (2015)), benefit from the high spatial resolution of these instruments (about 1 km).

OLCI (Ocean and Land Color Instrument) is a medium resolution imaging spectrometer, operating in the solar reflective spectral range (400 nm to 1040 nm). Two OLCI instruments, aboard Sentinel-3A (launched 2016) and -3B (launched 2018) satellites, are currently operational. The primary mission of OLCI is the observation of sea and land surfaces, with secondary mission of providing information on atmospheric constituents. OLCI is based on the design of MERIS, and provides continuity with MERIS with enhanced capabilities. For detailed description of Sentinel-3 mission and OLCI instrument, see Donlon et al. (2012). OLCI level 2 IWV product for land (OL_2_LFR/OL_2_LRR) builds on heritage of water vapour algorithm designed for MERIS instruments with similar differential absorption technique.

Between January 2020 and December 2021, OLCI/Sentinel-3 IWV (included in OL_2_LFR products) was validated within the "ESA/Copernicus Space Component Validation for Land Surface Temperature, Aerosol Optical Depth and Water Vapour Sentinel-3 Products" project (referenced in the following by LAW). The aim of the project was to perform more extensive and systematic validation against ground-based measurements of the following Sentinel 3 core products: the Integrated Water Vapour included in OL_2_LFR products, Aerosol Optical Thickness included in SY_2_AOD products and Land Surface Temperature provided by SL_2_LST products.

This paper is dedicated to validation of OLCI total column water vapor data. The paper is structured in the following way: Section 2 provides a brief description of the algorithm used in OLCI IWV retrieval with emphasis on the features relevant to validation work. Section 3 introduces the reference data sources used. Section 4 describes the match-up database generated as a part of LAW project, as well as the co-location criteria and screening applied to match-ups. Validation results and the discussion of results are shown in the Section 5 with conclusions in Section 6.

## 2   OLCI Integrated Water Vapour retrieval

OLCI instruments are currently operational aboard two Sentinel-3 satellites, orbiting the Earth on sun-synchronous polar orbits with equator crossing time at 10:00 local time. With swath width of 1270 km, two OLCI instruments provide revisit time of less than two days at the equator. Local times covered by OLCI observations wary from around 9:00 at the western edge of the swath to around 10:30 at the eastern edge of the swath. Horizontal resolution is about 300 m for full resolution products and 1200 m for the reduced resolution products. Spectral range of the OLCI instrument is 400-1040 nm, divided to 21 programmable spectral bands.

Total column water vapour (also labeled as integrated water vapour IWV) for cloud-free pixels is included in OLCI level 2 products for land (full resolution OL_2_LFR and reduced resolution OL_2_LRR) and water (OL_2_WFR/OL_2_WRR). OLCI level 2 products, including the IWV, are generated using a common pre-processing and product formatting process. Water vapour retrieval is performed during the common preprocessing step, as a part pixel classification process. Retrieval is based on the differential absorption technique using spectral radiances at water vapour absorption band at 900 nm and nearby water vapour reference band at 885 nm. The algorithm builds on the heritage of the retrieval algorithm designed for OLCI's precursor MERIS (Rast et al., 1999; Lindstrot et al., 2012). The water vapour column above a pixel is estimated by comparing Radiative Transfer (RT) based simulations with the corresponding OLCI measurements. The RT-simulations are approximated by a product of the atmospheric transmission (using exponential sums of pre-calculated uncorrelated k- distribution terms, (Doppler et al., 2013)) and an estimation of the scattering—absorption—interaction, quantified by a factor and stored in a look-up table (LUT). The optimisation with respect to the total column water vapour is done by a one-dimensional gradient descent (see also https://sentinel.esa.int/web/sentinel/technical-guides/sentinel-3-olci/level-2/water-vapour-retrieval, accessed on 21 June 2022).

Cloudy pixels are detected using standard OLCI level 2 cloud mask, which includes cloud ambiguous and cloud margin flags (see also https://sentinel.esa.int/web/sentinel/technical-guides/sentinel-3-olci/level-2/pixel-classification, accessed on 7 March 2022). For general overview of the OLCI instrument and products, see OLCI user handbook (https://sentinels.copernicus.eu/web/sentinel/user-guides/sentinel-3-olci, accessed on 7 March 2022).

Previous study using regional GNSS observations as reference indicates OLCI uncertainty within specifications (Mertikas et al., 2020). The quality of IWV product, as well as other OLCI land products, are also assessed on a monthly basis in OLCI Data Product Quality Reports (available online at https://sentinels.copernicus.eu/web/sentinel/technical-guides/sentinel-3-olci/data-quality-reports, accessed on 21 June 2022).

## 3    Reference data sources

### 3.1    IGRA radiosoundings

The Integrated Global Radiosonde Archive (IGRA) consists of quality-controlled radiosonde and pilot balloon observations from more than 2,800 globally distributed stations, of which about 800 are currently reporting data. Version 2 of the IGRA (Durre et al., 2016) includes new data sources and quality control procedures, as well as new user-requested variables. Version 2 also includes several derived parameters, including the precipitable water vapour between surface and 500 hPa pressure level, used in this study as the reference water vapour parameter. Description of the network and the quality-control measures applied can be found in Durre et al. (2018). The choice of limiting the water vapour column to 500 hPa pressure level leads to a dry bias, reported as 2.44% by Wang et al. (2007). However, it also avoids problems with the decreasing accuracy in colder temperatures of the upper troposphere (Van Malderen et al., 2014).

Soundings in IGRA database come from several sounding networks, using different radiosonde types with different processing. Due to this inhomogeneous nature, the independent uncertainty of the observation varies between stations. Wang and Zhang (2008) reports biases of around 1 kg/m$^2$ for different sonde types, with dry bias for capacitive polymer sondes and wet bias for carbon hygristor and Goldbeater's skin hygrometers. Estimated precision of sonde-based total columns compared to ground-based has been reported to be around 5% (Van Malderen et al., 2014).

The effective spatial and temporal resolution of the IWV from the radiosondes is affected by the drift during the radiosonde ascent and the time the sonde takes to ascend to required altitude. The median horizontal drift of the sonde during ascent to 500 hPa pressure level is about 10 km, although the extent of drift for individual observations can be much higher. The median drift has a latitudinal dependency with maxima at mid-latitudes. The median time of the ascent to 500 hPa level is 30 minutes (Seidel et al., 2011).

### 3.2    SUOMINET GNSS network

Long-term TCWV data sets from GNSS networks are widely used in studies involving atmospheric water vapour columns. In this study, we use U.S. SuomiNet (UCAR/COSMIC) TCWV product, which consists of observations of over 400 Global Positioning System (GPS) stations with near-global distribution (Ware et al., 2000). This large network provides TCWV values retrieved from consistently processed GPS measurements of the temperature- and humidity- dependent zenith path delay at a couple of hundred sites with a temporal resolution of 30 min (https://www.cosmic.ucar.edu/what-we-do/suominet-weather-precipitation-data). Analysis method and the data set is described in detail by Wang et al. (2007). Unlike radiosonde observations, which can be considered in-situ observations, displaced by wind during the ascent, GPS observations can be considered to represent a cone, covering an area of about 100 km$^2$ (Van Malderen et al., 2014).

Van Malderen et al. (2014) provides an overview of the uncertainties of the GPS-based IWV observations. They consider uncertainty from three parts of the retrieval: (1) Zenith Total Delay (ZTD) estimation, (2) zenith hydrostatic delay (ZHD) modeling, and (3) conversion of zenith wet delay (ZWD) to IWV. Formal error provided with the SUOMINET observations gives an estimate of the random uncertainty due to the first of these error sources (ZTD estimation), which is generally the

main source of uncertainty in IWV.(Deblonde et al., 2005) Uncertainties in ZHD modeling are due to errors in pressure measurements and can lead to IWV uncertainties of roughly 0.5 kg/m$^2$ (Deblonde et al., 2005; Wang et al., 2007). The main source of uncertainty in the conversion of ZWD to IWV is the mean atmospheric temperature. Estimated uncertainty of the mean atmospheric temperature, when calculated from surface temperature, is around 5K, which leads to IWV error between 0.07 (dry atmosphere) and 0.72 kg/m$^2$ (moist atmosphere) (Deblonde et al., 2005). Based on these three sources, uncertainty of IWV measurements from GPS observations can generally be considered to be less than 2 kg/m$^2$ (Van Malderen et al., 2014).

## 4 Match-up database and data selection

### 4.1 LAW match-up database

As a part of the LAW project, ACRI-ST created a database of matchups, gathering a combination of reference measurements (IGRA and SUOMINET) and satellite macro-pixel collocated in time and space. The matchup database was used as a basis of this work, and it is available upon subscription from LAW project web portal (https://law.acri-st.fr/home). In addition to IWV matchups, database includes matchups for OLCI Aerosol Optical Depth and Land Surface Temperature observations.

For IWV the database includes a matchup for each overpass over each reference station. For each match-up an OLCI pixel overlapping the reference station is included, as well as a macro-pixel of 31 × 31 OLCI pixels (i.e. a surface area of around 10 × 10 km) surrounding the reference station. All reference observations within a time window of +/- 3 hours of the overpass are included in the database. Satellite overpasses are generated to the database even when the OLCI IWV observation over the reference station is unavailable due to cloud contamination or retrieval failure. In these cases, the OLCI IWV observations are flagged with CLOUD or WV_FAIL quality flags, respectively. Satellite overpasses are only filtered from the database in the case of data lost due to operational issues or radio frequency interference (RFI) contamination from other satellites. Satellite extractions include quality flags and contextual parameters present in the Sentinel-3 operational products.

For the analysis presented here, matchups included in the database were further screened as detailed in the following section. Locations of screened IGRA and SUOMINET matchups with OLCI observations for OLCI/Sentinel-3A are shown in the Figure 1. Locations of matchups for OLCI/Sentinel-3B are similar (not shown).

### 4.2 Quality screening and data selection

In this study, only OLCI observations located over land (LAND flag) were considered. The IWV matchups from the LAW matchup database were further screened for failed OLCI inversion (WV_FAIL flag) and for cloud conditions. For cloud screening, observations with cloud warning flag (CLOUD) or with warning flags for possible cloud margin (CLOUD_MARGIN) and for ambiguous cloud conditions (CLOUD_AMBIGUOUS) were discarded. For each overpass, the satellite-reference observation pair with smallest time difference was chosen. For most of the analysis presented here, only OLCI pixels directly over the reference stations were used. The surrounding 31 × 31 pixels in the macropixel stored in the LAW database were only used in the validation of uncertainty estimates (See section 5.3).

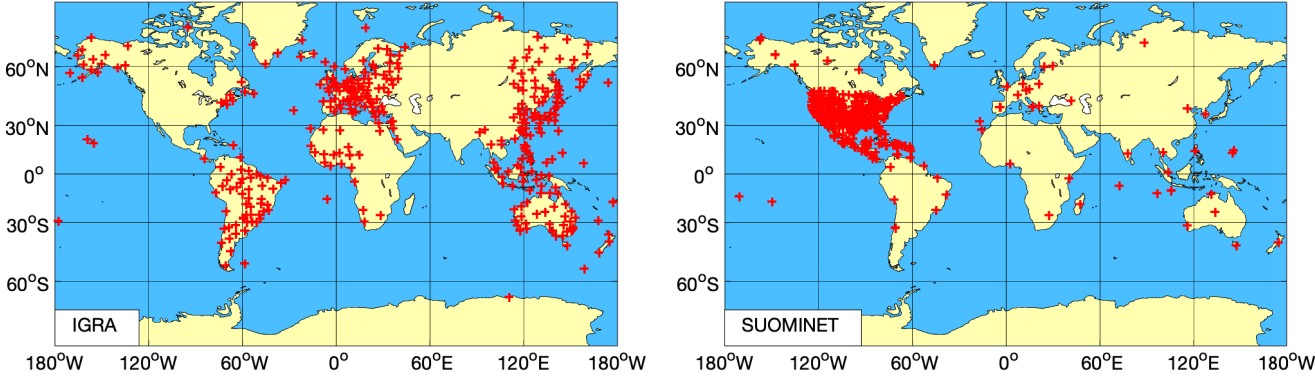

**Figure 1.** Locations of OLCI-A matchups with IGRA (left) and SUOMINET (right) networks.

Selection criteria were then applied to the matchups collected to the database based on the reference observation: For SUOMINET observations, the maximum time difference allowed was 15 minutes. The 15-minute limit was selected to preserve the 30 minute temporal resolution of the SUOMINET network. SUOMINET matchups were additionally required to have a reported SUOMINET formal error of less than 2 kg/m$^2$. For IGRA, time difference of up to 180 minutes was allowed between the satellite overpass and the radiosonde launch. Larger time differerence was used for radiosoundings because of low temporal sampling (typically twice per day) of the observations. The selection criteria were similar to those previously used by Kalakoski et al. (2016). Due to the fine spatial resolution of the OLCI instrument and the close spatial co-location criteria used, inhomogeneous terrain around coastal or mountainous reference stations is not expected to have a large effect on the validation results.

## 5 Results and Discussion

### 5.1 Overall agreement

Comparisons were carried out separately for each instrument (OLCI-A and -B) and for each reference dataset (SUOMINET and IGRA). Results of the general comparisons are presented in Figure 2 and Table 1. Agreement is generally good, with very high correlation coefficients (0.98 and 0.90 for SUOMINET and IGRA, respectively). The dispersal of the differences is considerably higher for IGRA matchups, partly due to longer time differences allowed, and the drift of the sondes during the ascent. Higher mean differences observed for IGRA comparisons can also partially be explained by the dry bias of lower tropospheric (500 hPa) column used in this study as a radiosonde reference value (Wang et al. (2007)).

All comparisons show a positive (wet) median bias for the OLCI observations, increasing linearly with increasing total water vapour content (Figures 2 and 3). For total water contents larger than 50 kg/m$^2$, the bias in IGRA matchups dips closer

to the zero line. As the similar dip is not observed in SUOMINET comparisons, the lower bias is likely related to uncertainties in radiosonde data or to relatively lenient collocation criteria used. General comparisons also indicate very good agreement between OLCI-A and -B. If observations flagged CLOUD_MARGIN and CLOUD_AMBIGUOUS are allowed, dispersal is much larger, with large number of outliers (not shown here). However, investigation of matchups with cloud-flagged pixels in the vicinity (See figure A1) indicates that the bias is not significantly affected by the nearby cloud-contaminated pixels. This indicates that the current cloud-flagging provides robust cloud-screening, with little cloud-induced uncertainty.

A dependencies of observed biases on latitude, solar zenith angle and season were also investigated (Figures 4 and 5). Results are consistent with the linear increase of wet bias seen in Figures 2 and 3. The dependency observed for latitude and and solar zenith angle is likely related to generally higher water vapour total columns typical of low latitudes and solar zenith angles. As noted above, the drift statistics of the radionsondes also show a latitudinal variation, possibly affecting the dispersal at certain latitudes.

**Table 1.** Statistics of general comparisons shown in figure 2.

| Parameter | SUOMINET | | IGRA | |
|---|---|---|---|---|
| | OLCI-A | OLCI-B | OLCI-A | OLCI-B |
| Number of co-locations | 46758 | 49078 | 20708 | 21021 |
| Mean difference [kg/m$^2$] | 1.71 | 1.63 | 2.58 | 2.59 |
| Standard deviation of difference [kg/m$^2$] | 2.93 | 3.07 | 6.22 | 6.41 |
| Mean relative difference [%] | 10.6 | 10.1 | 15.6 | 16.2 |
| Standard deviation of relative difference [%] | 21.3 | 22.2 | 44.8 | 49.8 |
| Correlation coefficient | 0.98 | 0.98 | 0.90 | 0.90 |

## 5.2  Anomaly in OLCI-B camera 3

While in general a validation results of OLCI-A and -B are very similar, a small anomaly in distribution of differences was observed in SUOMINET comparisons for OLCI-B camera 3, compared to the distribution for the other cameras in either instrument (Figure 6, top panels). After separating the differences by the central wavelength of the relevant instruments (Figure 6, middle and bottom panels), the anomaly in OLCI-B Camera 3 (Bottom panel, yellow line) was observed at all wavelengths, clearly distinguishable from the other cameras. This points to a conclusion that the anomaly is not due to known differential drift in camera 3, but rather due to an uncorrected instrumental issue. For more information on OLCI spectral characterization and drift of the central wavelengths, see the Sentinel Online website (https://sentinels.copernicus.eu/web/sentinel/technical-guides/ sentinel-3-olci/olci-instrument/spectral-characterisation-data, accessed 3 March 2022), and technical note available at the website. A small left-right bias is also seen between camera 1-5 in both instruments. This could be due to the difference in local time (about 45 minutes between cameras 1 and 5) and consequently observed total columns. Neither the left right difference or the anomaly in OLCI-B / camera 3 is observed in IGRA comparisons, due to larger dispersal of the differences.

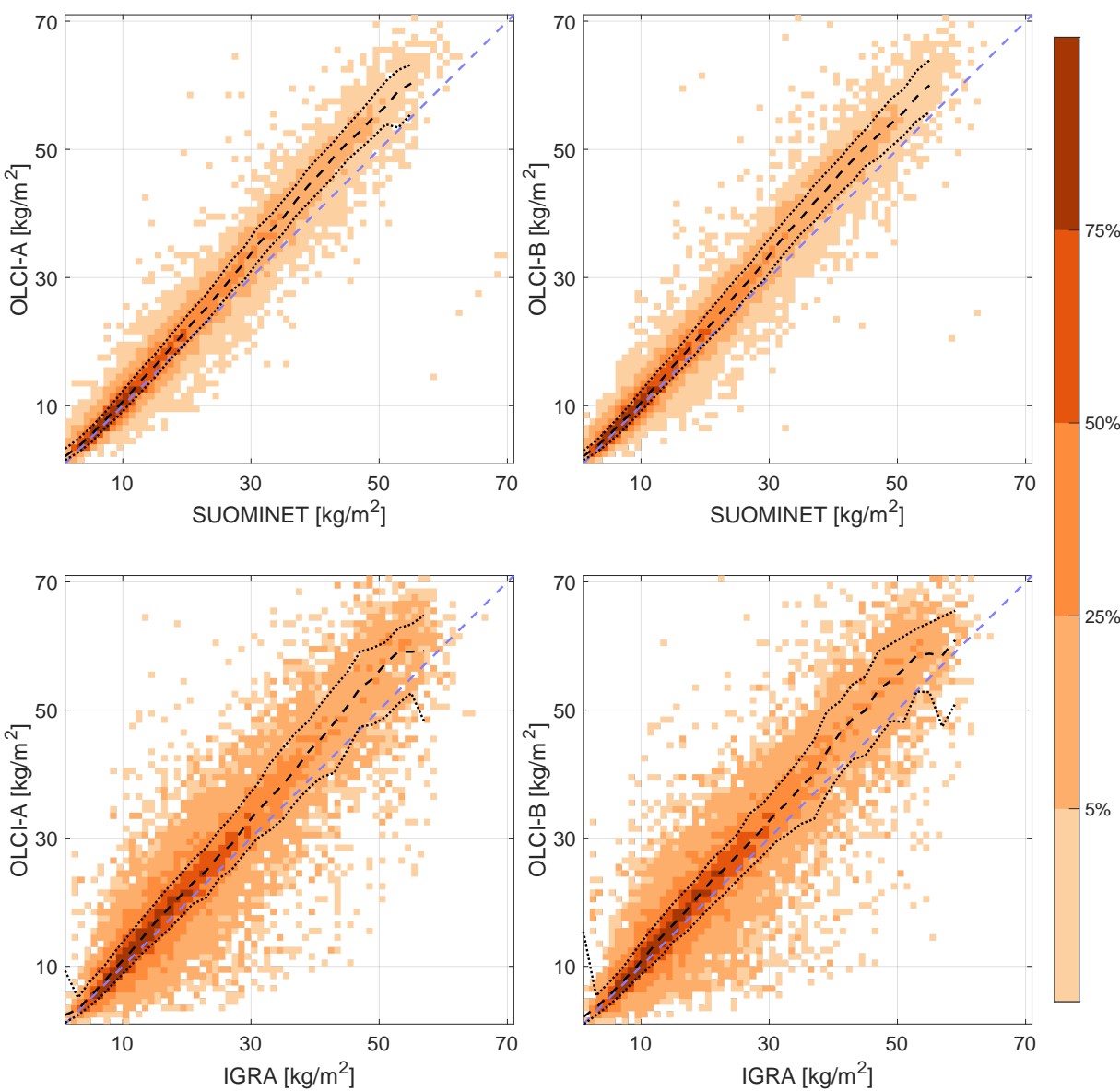

**Figure 2.** Density scatter plot of LAND comparisons of OLCI-A (left) and -B (right) against SUOMINET (top) and IGRA (bottom) observations, with CLOUD_MARGIN and CLOUD_AMBIGUOUS matchups removed. Color field shows the percentage of matchups in each category, with darkest colours showing the highest density of matchups. Blue dashed line shows the x = y line and the black lines median (dashed) and 16th and 84th percentiles (dotted) OLCI-A observation for each 2 kg/m$^2$ bin of reference observations. Linear fits of the matchups (not shown) for SUOMINET are y = 1.12x-0.31 (OLCI-A) and y = 1.11x-0.29 (OLCI-B) and for IGRA y = 1.07x+0.69 (OLCI-A) and y = 1.07x+0.64 (OLCI-B). Correlation coefficients are 0.98 (SUOMINET) and 0.90 (IGRA) for both instruments.

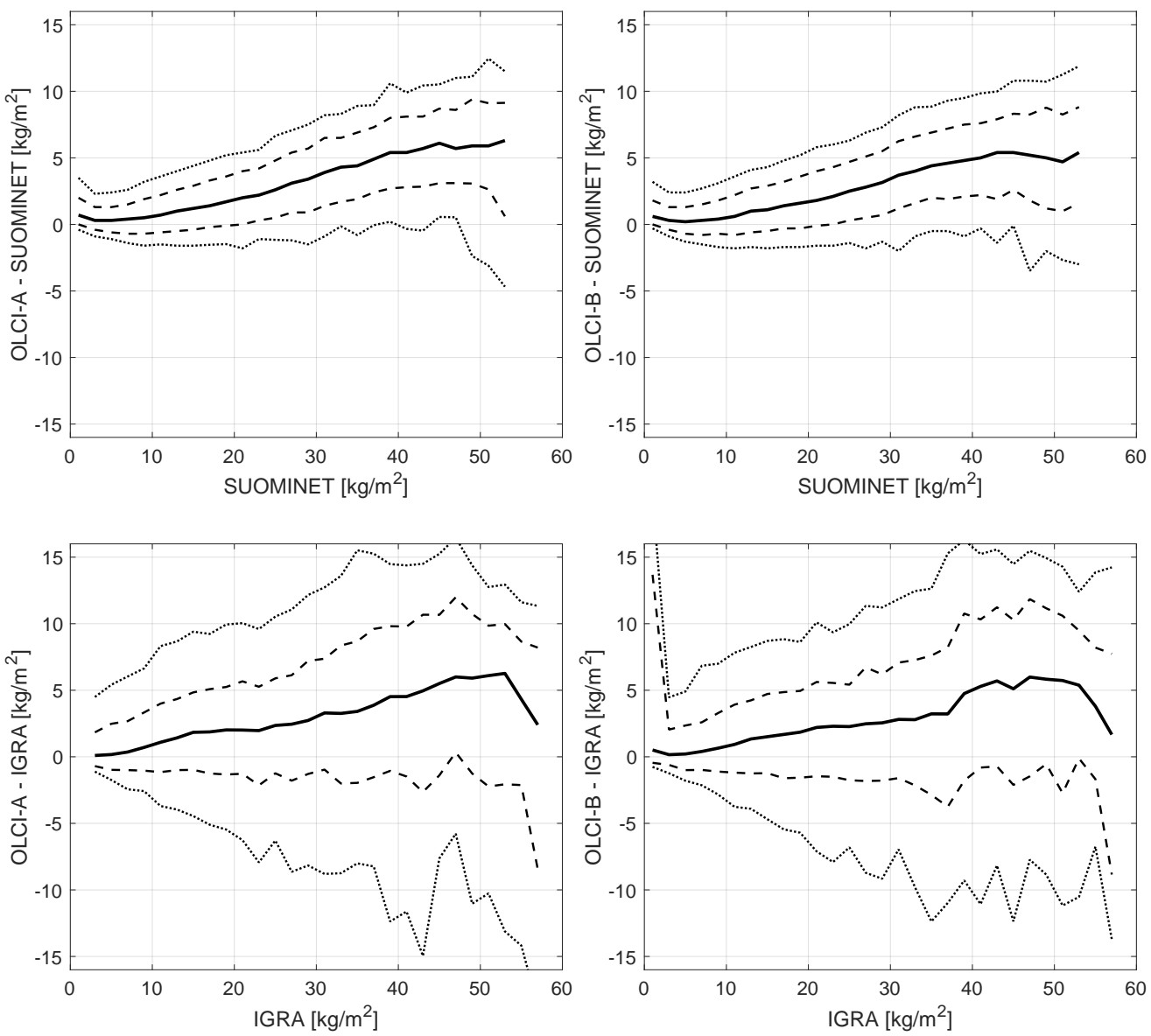

**Figure 3.** Difference OLCI of observations against SUOMINET (top) and IGRA (bottom) for OLCI-A (left) and OLCI-B (right). Solid line shows the median of each 2 kg/m$^2$ wide bin, while the dashed show the 16th and 84th percentiles and the dotted lines the 5th and 95th percentiles. Bins with less than 20 matchups were omitted from the figure.

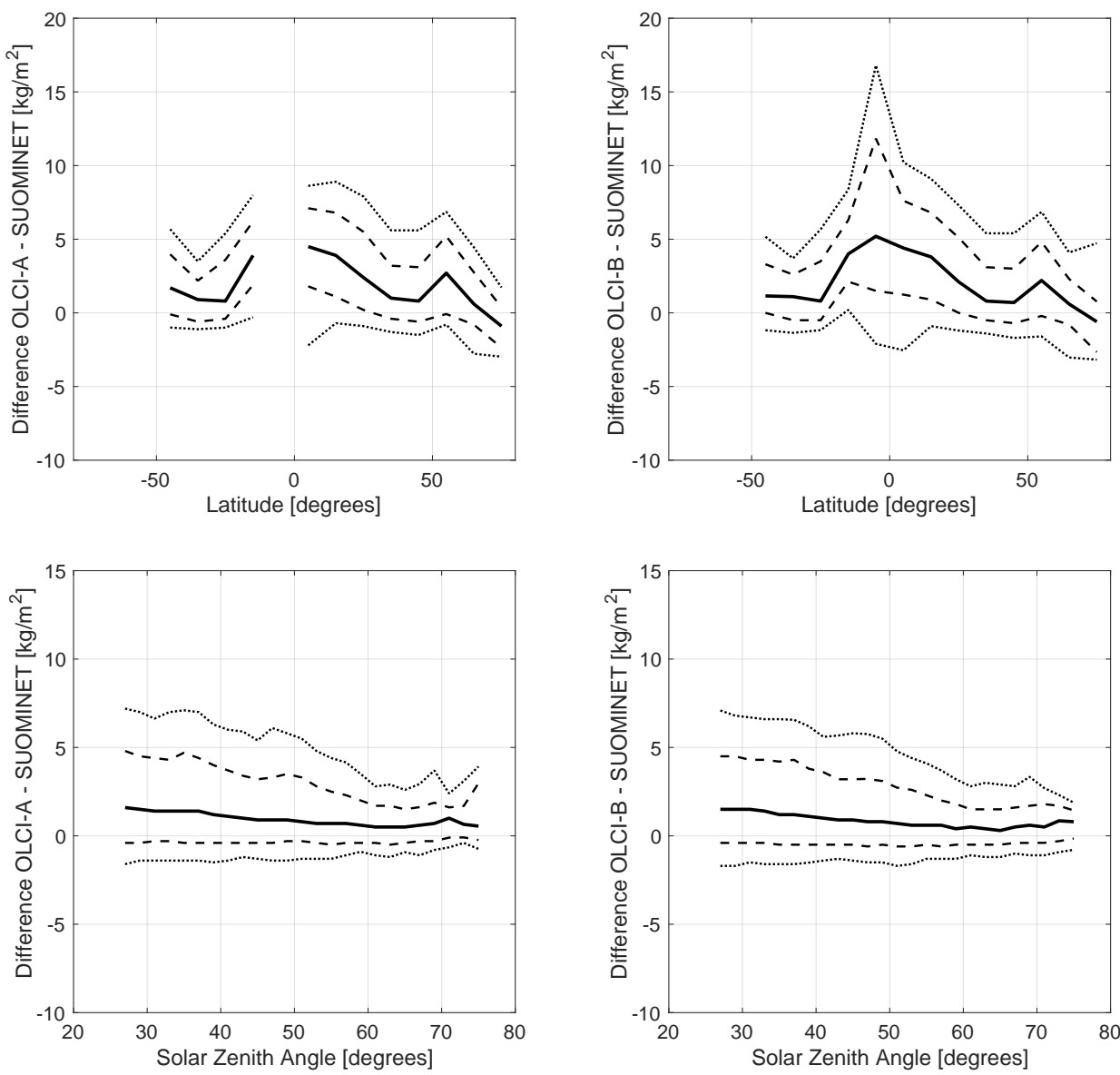

**Figure 4.** Difference OLCI-A (left) and -B (right) of observations against SUOMINET as a function of latitude (top row) and Solar zenith angle (bottom row). Solid line shows the median of each 2 degree wide bin, while the dashed show the 16th and 84th percentiles and the dotted lines the 5th and 95th percentiles. Bins with less than 20 matchups were omitted from the figure.

## 5.3 Validation of error estimates

For the validation of random uncertainty estimates, we use the structure function method described in detail in Sofieva et al. (2021). This method is based on evaluation of the structure function, i.e., root-mean-square differences as a function of in-

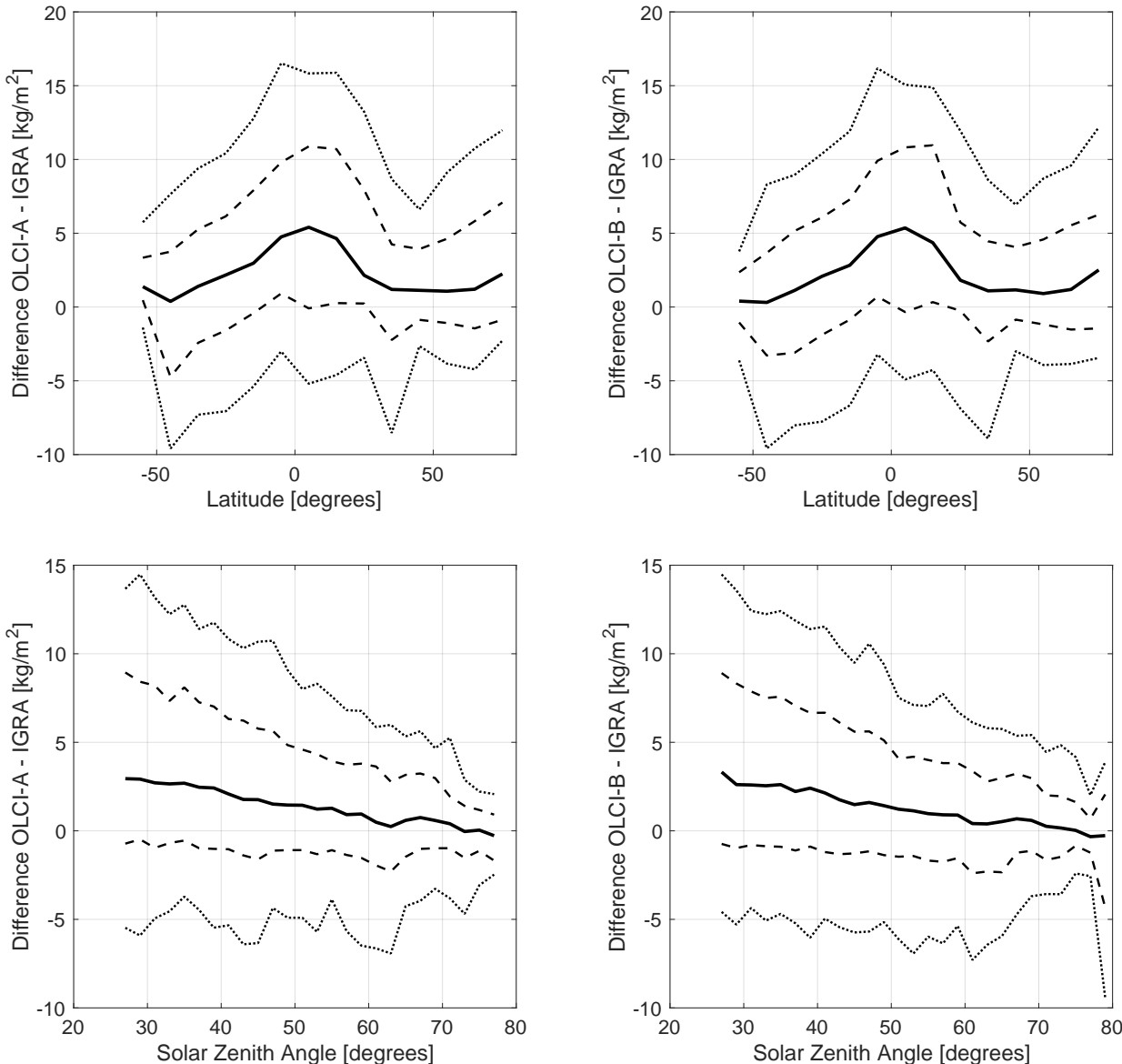

**Figure 5.** Difference OLCI-A (left) and -B (right) of observations against IGRA as a function of latitude (top row) and Solar zenith angle (bottom row). Solid line shows the median of each 2 degree wide bin, while the dashed show the 16th and 84th percentiles and the dotted lines the 5th and 95th percentiles. Bins with less than 20 matchups were omitted from the figure.

creasing spatio-temporal separation of the measurements. The limit at the zero mismatch provides the experimental estimate of random noise in the data. For the analysis shown here, we used the OLCI-A and -B data from the cloud-free SUOMINET matchups over land.

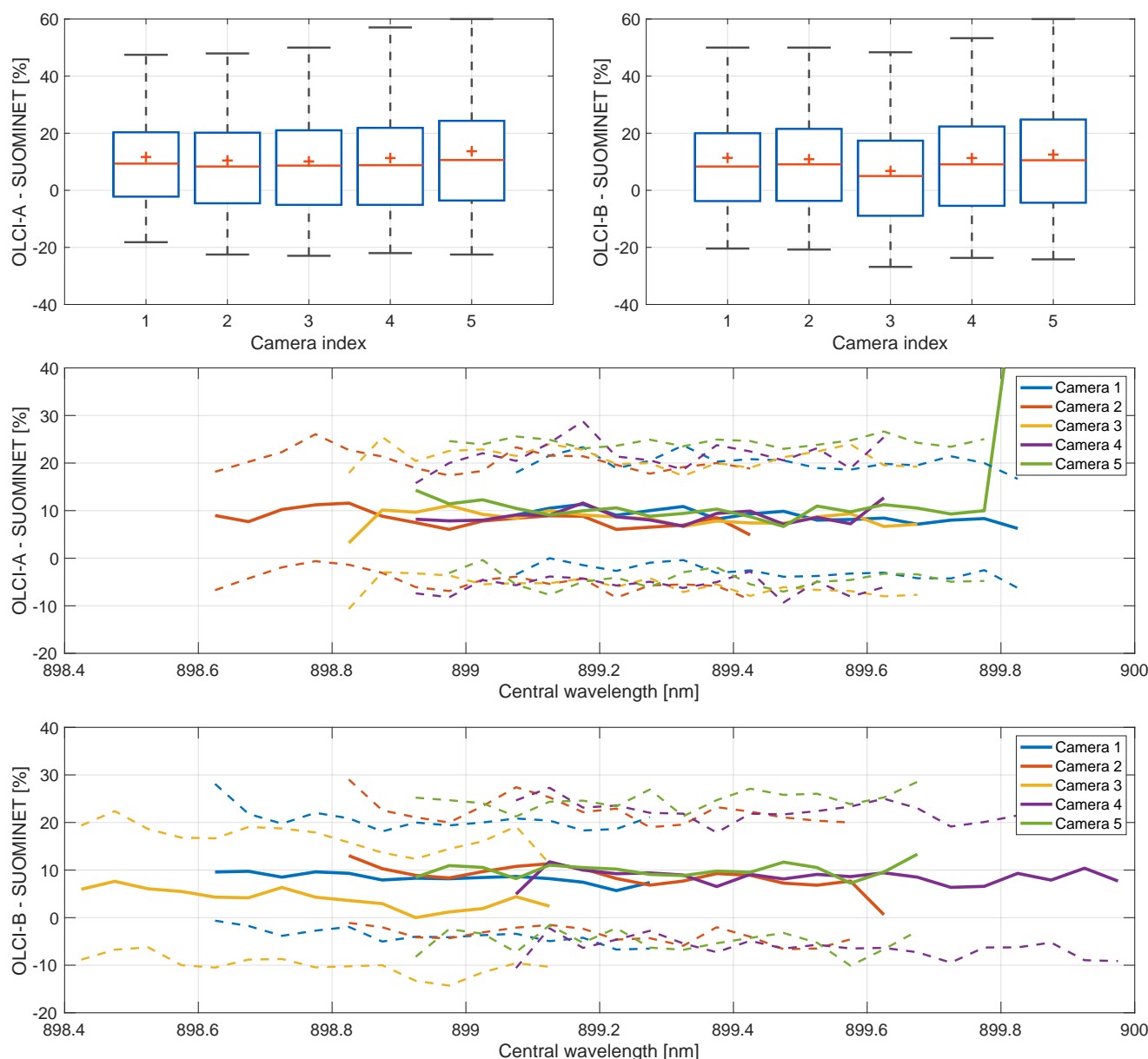

**Figure 6.** Dependency of OLCI - SUOMINET relative difference on camera index. Top row: Distributions of relative differences for OLCI-A (left) and OLCI-B (right) match-ups with SUOMINET observations for OLCI cameras 1-5. Red lines show the medians of the distributions and blue boxes the 16th and 84th percentiles of the distribution with black whiskers showing the 2.5th and 97.5th percentiles. Red crosses show the means of the distributions. Middle and bottom rows: Relative difference as a function of central wavelength of the detector for OLCI-A (middle panel) and OLCI-B (bottom panel). Colors represent cameras 1-5 (see legend) and the solid and dashed lines represent median and the interpercentile range (16th-84th) percentiles, respectively.

In order to validate the error estimates provided by the OLCI IWV algorithm, we investigated the difference of OLCI observations within the $31 \times 31$ pixel OLCI macropixel to the center pixel, and computed sample variances. RMS difference increases as a function of the distance from center (Figure 7, top-left panels). For comparison, the bottom-left panels of 7 show the mean error estimate from the OLCI product. The mean of structure function for the eight pixels around the center pixel

was taken to represent the experimental uncertainty estimate for the OLCI IWV. Right panels of figure 7 show the distributions of the experimental uncertainty estimates and the estimates given by the OLCI algorithm. The distributions of the estimates overlap, showing that the two estimates are consistent with each other. Experimental estimates are generally lower than the ones provided by the algorithm. This is partly caused by the $0.3$ kg/m$^2$ increments of OLCI error estimates, which reduces the sensitivity of the OLCI estimate, especially at the lower end of the distribution. In general, the validation performed confirms

the validity of the provided error estimates. However, the quantization is too coarse to provide accurate random uncertainty estimates.

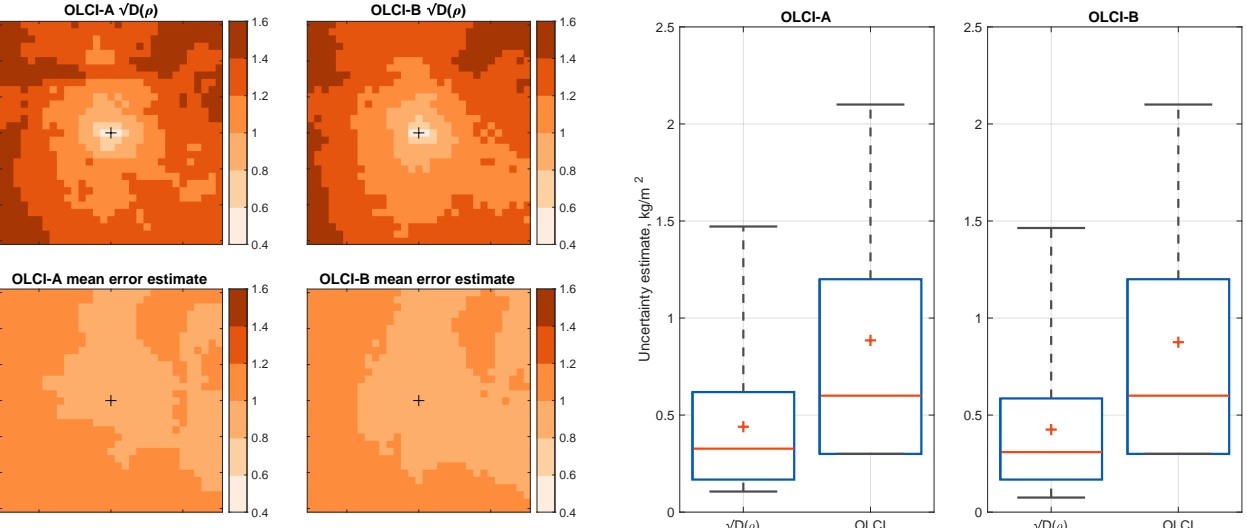

**Figure 7.** Left panels: Structure function ($\sqrt{D(\rho)}$ in kg/m$^2$) within macropixel, compared to center of the macropixel (Top) and the square root of mean error estimate of corresponding OLCI pixels (bottom). Rigth panels: Distribution of experimental uncertainty estimates (8 pixels around the center of the macropixel) using the structure function method $\sqrt{D(\rho)}$ and the IWV uncertainty estimates from inversion algorithm. Red lines show the medians of the distributions and blue boxes the 16th and 84th percentiles of the distribution with black whiskers showing the 5th and 95th percentiles. Red crosses show the means of the distributions.

## 6   Conclusions

OLCI IWV was validated against two reference datasets, SUOMINET GNSS observations and IGRA integrated radiosonde columns. High correlation with the reference observations (0.98 for SUOMINET and 0.90 for IGRA) was observed for both OLCI-A and -B, with all comparisons showing a wet bias of 7-10%. Notably, the results of the general comparisons were very similar for OLCI-A and B.

In more detailed comparisons, wavelength dependency of an observed anomaly in OLCI-B/camera 3 was investigated, showing that the anomaly is independent of the central wavelength of the relevant sensor. This indicates that the cause of the anomaly is not the wavelength drift of the sensors in camera 3. Proximity of clouds within the macropixel was shown to have little effect on the observed differences, confirming the robustness of cloud-flags provided with the OLCI product.

Error estimates of the OLCI product were compared to an experimental estimate of random uncertainty. Comparisons indicate that the OLCI estimates were consistent with the experimental estimates, but generally higher. This is partly due to large increment ($0.3$ kg/m$^2$) of the reported OLCI error estimates.

As an outcome of the validation work carried out within the LAW project, three main recommendations were submitted: 1.) Possibility of reducing the wet bias using additional OLCI channels (see Preusker et al., 2021) should be investigated, 2.) correction to the anomaly observed in Camera 3 / OLCI-B should be implemented, and 3.) uncertainty estimates should be revisited, preferably with smaller increments for better characterization.

*Data availability.* Matchup data base containing the data used for this work is available on subscription at https://law.acri-st.fr/home

*Author contributions.* N.K. and V.S. planned the validation approach. C.H. and M.D. produced the matchup database. N.K. performed the analysis and prepared the manuscript with contributions from all authors.

*Competing interests.* The authors declare that they have no conflict of interest.

*Acknowledgements.* The work presented here was funded by "ESA/Copernicus Space Component Validation for Land Surface Temperature, Aerosol Optical Depth and Water Vapour Sentinel-3 Products" project.

Authors would like to thank UCAR and Suominet project for the SUOMINET data and NOAA for the IGRA data.

FMI thanks the Academy of Finland (Centre of Excellence in Inverse Modelling and Imaging).

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

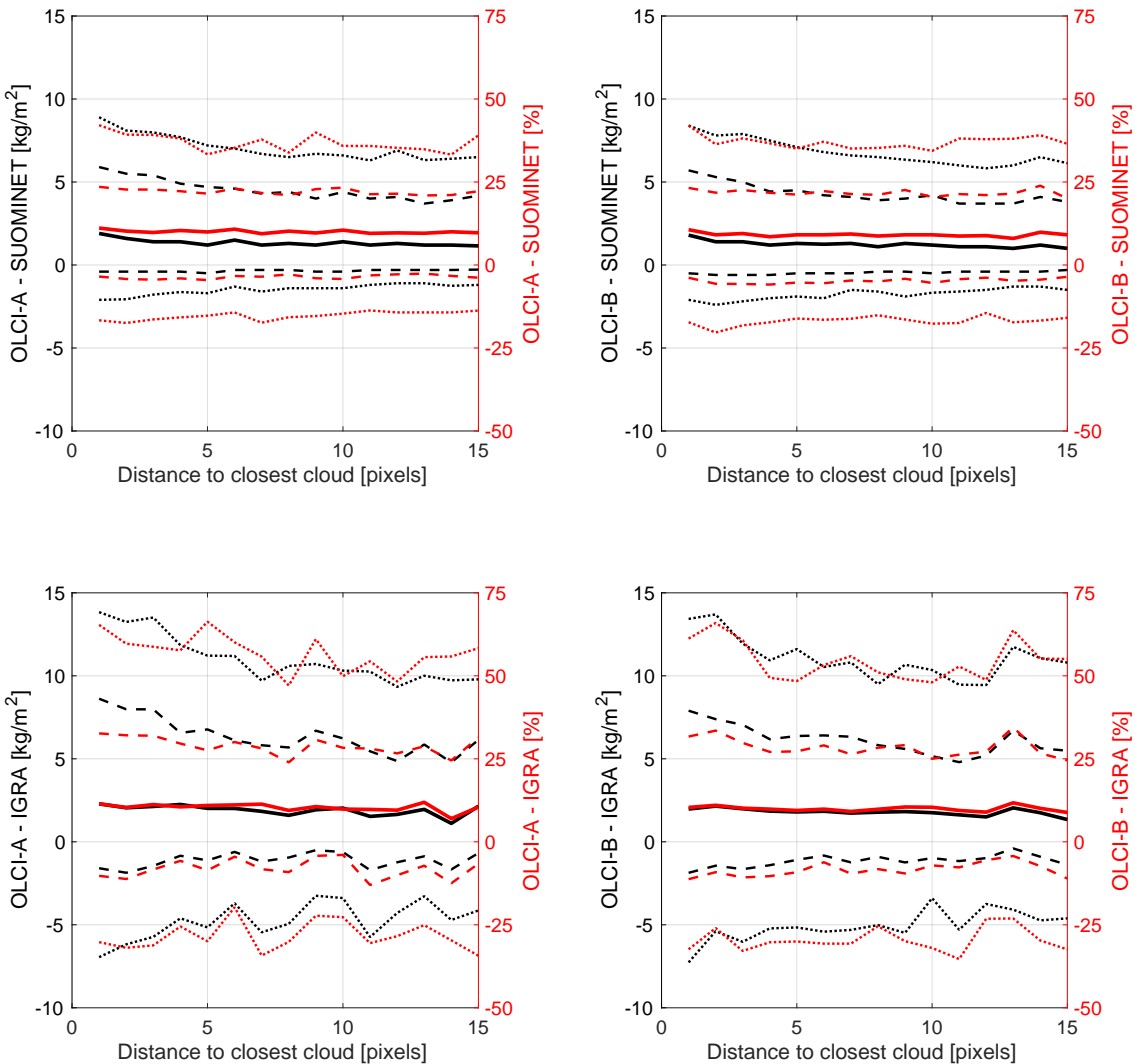

**Figure A1.** Dependency of observed OLCI difference against SUOMINET (Top) and IGRA (bottom) observations to distance to closest cloud-flagged pixel for OLCI-A and -B (left and right columns, respectively). Black lines show the difference in kg/m$^2$, while the red lines show the relative difference in percent. Solid line shows the median of each 2 kg/m$^2$ wide bin, while the dashed show the 16th and 84th percentiles and the dotted lines the 5th and 95th percentiles.