# Peer review of "Validation of Copernicus Sentinel-3/OLCI Level 2 LAND Integrated Water Vapour product"

_Atmospheric Measurement Techniques, 2022_

## Referee Comment (RC1)

The paper entitled 'Validation of Copernicus Sentinel-3/OLCI Level 2 LAND Integrated Water Vapour product by Niilo Kalakoski. Geophysical validation of the Integrated Water Vapour (IWV) from Sentinel-3 Ocean and Land Colour Instrument (OLCI) was performed as a part of "ESA/Copernicus Space Component Validation for Land Surface Temperature, Aerosol Optical Depth and Water Vapour Sentinel-3 Products" (LAW) project. The IWV was compared with reference observations from two networks: GNSS (Global Navigation Satellite System) derived precipitable water vapour from the SUOMINET network and integrated lower tropospheric columns from radio-soundings from the IGRA (Integrated Radiosonde Archive) database. The obtained results for cloud-free matchups overland with a high correlation against the reference observations SUOMINET as well as IGRA. Space based IWV have inherent uncertainties and need to be validated time to time basis before in operational or making the data in repository for future research. In that respect the present study have a high potential for publication after incorporation of the comments/suggestions as given below:

**MAJOR REVISIONS:**

1. **No discussion of other satellites that provide IWV in the introduction (e.g. MODIS, SCIAMACHY, GOME-2, AIRS).**

2. **Details about retrieval algorithm of Sentinel-3/OLCI Level 2 LAND Integrated Water Vapour product are missing and also give references i.e.**

i)      **How does Sentinel-3/OLCI provide LAND IWV data?**

ii)     **How is Sentinel-3/OLCI LAND IWV level2 data product generated?**

iii)    **How is Sentinel-3/OLCI LAND IWV level2 data different from radiosonde (IGRA) and GNSS in measuring/estimating IWV?**

iv)     **Are there any limitations of Sentinel-3/OLCI LAND IWV level2 data product based on former evaluation study (more literature reviews are needed).**

v)      **What is the horizontal resolution of IWV derived from Sentinel-3 OLCI LAND IWV level2 data?**

vi)      **Which method was used to identify cloud free pixels?**

3. **Line-76: Give references.**

4. **Section 3.1 & 3.2:** You have used Radiosonde & GNSS data as reference for comparison with **OLCI Level 2 LAND IWV**. But the Radiosonde & GNSS based data also associated with errors. Explain the possible sources of errors in your analysis with references.

**5. For IWV matchups a macro pixel of 31 × 31 OLCI pixels (i.e. a surface of around 10 by 10 km) with central pixel over each reference station is extracted at each overpass. What is horizontal resolution of OLCI IWV products and why you have selected 31x31 pixels? Have you applied any interpolation technique for resampling of OLCI IWV data over reference?**

**Specific Comments:**

6. All ground-based measurements acquired in a time window of +/- 3 hours are considered.

   It is not clear here in matchup criteria of time window of +/- 3 hours you considering for radiosonde (IGRA) or GNSS. Kindly mention time window for radiosonde (IGRA) for consideration of data along with which UTC data have been utilized for this study and same for GNSS also.

7. Missing satellite observations were only filtered from the database in the case of operational issues or radio frequency interference (RFI) contamination.

   How radio frequency interference (RFI) contamination occur in your data?

8. For all matchups, we applied an additional quality check according to quality flags. The matchups with failed inversion (WV_FAIL flag set) or with cloud warning flag (CLOUD), were discarded.

   For this study, you have chosen data from Sentinel-3 OLCI LAND IWV during cloud free pixel only then why again applied additional quality check. Please clarify.

9. For each matchup, the satellite-reference observation pair with smallest time difference was chosen. For SUOMINET, matchups with time differences larger than 15 minutes, or nominal error larger than 2 kg/m2 were not used.

   Kindly give the references.

10. The dispersal of the differences is considerably higher for IGRA matchups, partly due to longer time differences allowed, and the drift of the sondes during the ascent.

    Higher differences may due to the radiosonde ascents drift and vertical extent will be different over different geographical domains. Similarly, the collocations matchups of clear sky pixel retrievals will vary and hence the differences values also vary latitudinal.

11. Observed in SUOMINET comparisons, the bias reduction can be related to radiosonde data or to collocation criteria. General comparisons also indicate very good agreement between OLCI-A and -B.

What is retrieval algorithm of IWV of OLCI-A and -B. How bias can be reduce and related to radiosonde data.

12. Line120: INLAND_WATER water pixels, representing rivers and lakes, similarly show wet bias and large dispersal.

Give references.

13. The dependency observed for latitude and solar zenith angle is related to generally higher water vapour total columns seen in low latitudes and solar zenith angles,while the seasonal cycle is consistent with the over-presentation of northern hemisphere stations and higher total columns during summer months.

Give some prove or reference for this claim?

14. Line 161-168: OLCI observations classified as water surfaces (WATER and INLAND flags, including TIDAL with WATER) considerably larger bias and dispersal than those classified as land surfaces (LAND 165 flag, including TIDAL with LAND).

OLCI observations in pixels data contains both sea and mountainous land together along with topographically diverse terrains around these stations may introduce large bias.

**Technical Corrections:**

Line 90: May need prove reading carefully.

---

## Referee Comment (RC2)

**Comments on "Validation of Copernicus Sentinel-3/OLCI Level 2 LAND Integrated Water Vapour product" by Niilo Kalakoski et al.**

Anonymous reviewer
22 April 2022

The paper presents the analysis of validation of the Integrated Water Vapour (IWV) from Sentinel-3 Ocean and Land Colour Instrument (OLCI). The statistics is based on the comparison with data from the SUOMINET network and from the IGRA database. In my opinion, the scientific value of this research is rather high since IWV is the atmospheric parameter required for solving a large number of problems of climate change and weather prediction. Besides, the scientific interest to the described results is determined by the fact that the presented analysis is comparative (the data are provided by three independent sources) and covers different geographical locations. The obtained results could be helpful for the improvement of the retrieval algorithm used to process the OLCI measurements.

Despite the fact that the paper contains valuable and interesting results, I can not recommend it for publication in the present form. **The paper requires major revision.** There are several issues of general character which should be addressed in this paper. Besides, there are also specific remarks.

**General critical issues:**

1) The IWV measurements by OLCI are not placed within the context of modern space-based observations of IWV. Such placing is especially important since providing information on atmospheric parameters is a secondary goal of the OLCI mission and not a primary goal as the authors indicate in the Introduction section. My recommendation is to give a comparison of the IWV retrieval accuracy values and ground pixel size values which are declared or estimated for OLCI and other satellite instruments which measure IWV. Such comparison would help to assess the value of the IWV observations by OLCI.

2) Section 2 "OLCI Integrated Water Vapour retrieval" is too sketchy. I recommend giving more details of the retrieval algorithm instead of giving only references to technical guidelines. In addition, full resolution and reduced resolution modes should be explained which are now only briefly mentioned (Page 2, line 46).

3) The description of the data selection procedure in Section 4 is very unclear. In particular, the sentence "Missing satellite extractions…" (Page 4, Line 88) can be misleading. Please explain what you mean when you say that the reference observations should be validated (Page 4, Lines 90-91). I recommend the authors to remove current subsections of Section 4 and to provide the information on data selection and quality control on step-by-step basis separately for IGRA and SUOMINET. All criteria for match-up should be given clearly. The reasons for choosing 31x31 macropixels should be presented also.

4) In my opinion, the presentation of the results in Section 5 requires improvement:

- The cloud-screening criteria (Page 4, Line 102) have been already described in Section 4. Mentioning these criteria in Section 5.1 can produce a wrong impression that they were used not in all cases.

- In order to avoid confusion, I recommend not to mention the results with WATER pixels. The authors note that they make validation for land pixels and "WATER pixels are not strictly part of the ESA OLCI product".

- I can not understand whether figures A1 and A2 belong to any Attachment or not. I would like to see these figures and relevant expanded discussion within the main text. I am not sure if bottom rows in these figures are necessary since all latitudes are taken together and therefore seasonal dependence seems to be hard to detect. Besides, I have the feeling that Fig. 3 duplicates information which is already contained in Fig. 2.

- The title of Subsection 5.2 "Classification of biases" is misleading because mainly the problem with camera 3 is discussed. I am not sure if Fig. 5 is necessary. It is sufficient to mention in the main text about negligible influence of neighbouring cloud-flagged pixels.

- In Subsection 5.3 please explain how the distribution of the error estimates from the retrieval algorithm was obtained. I suppose that every single OLCI measurement has its own error estimation and you just collected this information from all OLCI measurements. Is it so?

- I would like to recommend removing Subsection 5.4 and Fig. 7 in order to avoid confusion. As I have already mentioned above, the reason is that water pixels are not included in the OLCI products. If the authors decide to keep this subsection, all terms should be explained (WATER, TIDAL, INLAND), the number of observations in this critical pixels should be given, the observational conditions should be described (water, ice, ice covered with snow) etc.

**Specific remarks:**

Page 1, Line 2
What is the meaning of the term "geophysical" in the context of validation?

Page 1, Line 15
Remove the abbreviation ECV which is not used in the text below.

Section 3.1
Is it possible to roughly estimate the typical horizontal drift of a radiosonde during ascent to the pressure level of 500 hPa in order to assess the "effective" horizontal resolution of radiosonde measurements of IWV?

Section 3.2
If possible, please provide the information about pressure level which is assumed as an upper limit for IWV in the SUOMINET GNSS network? What is the area of horizontal averaging for the derivation of IWV by the GNSS method?

Page 4, Line 82
Please explain the acronym ACRI-ST.

Page 5, Line 112
Please compare typical time of the sonde ascents to the level of 500 hPA with the allowed time mismatch between OLCI and sonde observations.

---

## Author Comment (AC1)

Answer to comments from reviewer 1:

Kalakoski et. al.: Validation of Copernicus Sentinel-3/OLCI Level 2 LAND Integrated Water Vapour product

The paper entitled 'Validation of Copernicus Sentinel-3/OLCI Level 2 LAND Integrated Water Vapour product by Niilo Kalakoski. Geophysical validation of the Integrated Water Vapour (IWV) from Sentinel-3 Ocean and Land Colour Instrument (OLCI) was performed as a part of "ESA/Copernicus Space Component Validation for Land Surface Temperature, Aerosol Optical Depth and Water Vapour Sentinel-3 Products" (LAW) project. The IWV was compared with reference observations from two networks: GNSS (Global Navigation Satellite System) derived precipitable water vapour from the SUOMINET network and integrated lower tropospheric columns from radio-soundings from the IGRA (Integrated Radiosonde Archive) database. The obtained results for cloud-free matchups overland with a high correlation against the reference observations SUOMINET as well as IGRA. Space based IWV have inherent uncertainties and need to be validated time to time basis before in operational or making the data in repository for future research. In that respect the present study have a high potential for publication after incorporation of the comments/suggestions as given below:

The authors thank the reviewer for the careful reading and constructive comments. Several of the main issues raised by the reviewer were deliberate choices in order to keep the paper concise. However, we also appreciate that clarification and extension is needed in several places, notably the algorithm and data selection descriptions.

Please find below our answers (in blue) to the comments presented (in black).

**MAJOR REVISIONS:**

No discussion of other satellites that provide IWV in the introduction (e.g. MODIS, SCIAMACHY, GOME-2, AIRS).

Discussion of IWV from other satellite instruments extended. Extended discussion can be found on lines 22-42.

Details about retrieval algorithm of Sentinel-3/OLCI Level 2 LAND Integrated Water Vapour product are missing and also give references i.e.

    i)  How does Sentinel-3/OLCI provide LAND IWV data?

    ii)  How is Sentinel-3/OLCI LAND IWV level2 data product generated?

    iii)  How is Sentinel-3/OLCI LAND IWV level2 data different from radiosonde (IGRA) and GNSS in measuring/estimating IWV?

    iv)  Are there any limitations of Sentinel-3/OLCI LAND IWV level2 data product based on former evaluation study (more literature reviews are needed).

v) What is the horizontal resolution of IWV derived from Sentinel-3 OLCI LAND IWV level2 data?

vi) Which method was used to identify cloud free pixels?

Description of the retrieval algorithm in Section 2 was extended to consider these questions.

Line-76: Give references.

We added reference to SUOMINET website to following sentence and moved the Ware et. Al. reference to this sentence. See lines 110-114 of the revised manuscript.

Section 3.1 & 3.2: You have used Radiosonde & GNSS data as reference for comparison with OLCI Level 2 LAND IWV. But the Radiosonde & GNSS based data also associated with errors. Explain the possible sources of errors in your analysis with references.

Description of error sources was extended in Sections 3.1 and 3.2.

*For IWV matchups a macro pixel of 31 × 31 OLCI pixels (i.e. a surface of around 10 by 10 km) with central pixel over each reference station is extracted at each overpass.*

What is horizontal resolution of OLCI IWV products and why you have selected 31x31 pixels? Have you applied any interpolation technique for resampling of OLCI IWV data over reference?

Horizontal resolution of full resolution OLCI IWV is 300 metres. 31 x 31 pixel area was chosen as a compromise between acquiring enough values around the closest co-location and the storage requirements. No interpolation was applied to OLCI products in this study. Sections 4.1 and 4.2 were clarified in the revised accordingly.

Specific Comments:

*All ground-based measurements acquired in a time window of +/- 3 hours are considered.*

It is not clear here in matchup criteria of time window of +/- 3 hours you considering for radiosonde (IGRA) or GNSS. Kindly mention time window for radiosonde (IGRA) for consideration of data along with which UTC data have been utilized for this study and same for GNSS also.

The description of selection criteria was clarified. 3-hour time window was used for the matchups with IGRA observations to allow for more matchups to be generated from relatively limited number of soundings. For the SUOMINET observations the more stringent 15-minute criterion was applied. Vast majority of soundings used here are launched just before 00UTC or 12UTC.

*Missing satellite observations were only filtered from the database in the case of operational issues or radio frequency interference (RFI) contamination.* How radio frequency interference (RFI) contamination occur in your data?

RFI contamination can occur when other satellites interfere with the transmission of the Sentinel-3 observations to the ground station. In OLCI, this can lead to a loss of a few data packets creating data gaps over a few rows. RFI contamination can occur roughly 10 times a month, each time impacting a few lines. Sentence was clarified (lines 139-140).

*For all matchups, we applied an additional quality check according to quality flags. The matchups with failed inversion (WV_FAIL flag set) or with cloud warning flag (CLOUD), were discarded.* For this study, you have chosen data from Sentinel-3 OLCI LAND IWV during cloud free pixel only then why again applied additional quality check. Please clarify.

The screening for the CLOUD flag here is the part where we select the cloud-free observations. The observation is called as cloud free, when none of the cloud flags (CLOUD, CLOUD_MARGIN, CLOUD_AMBIGUOUS) are set. The text about the selection criteria clarified accordingly (lines 146-147).

9. *For each matchup, the satellite-reference observation pair with smallest time difference was chosen. For SUOMINET, matchups with time differences larger than 15 minutes, or nominal error larger than 2 kg/m2 were not used.* Kindly give the references.

The data selection criteria are the same as in Kalakoski et. al. 2016. We added this reference in the revised version of the paper.

The selection criteria for SUOMINET were largely selected based on personal experience. Maximum time difference of 15 minutes was selected to avoid observations affected by short-term data gaps. SUOMINET observations are normally available at 30 minute intervals, thus the 15 minute maximum difference ensures that the selected observation comes from an unbroken sequence. Nominal error limit was chosen based on analysis of the distribution of nominal error values. Neither requirement is very strong and as a consequence this screening removes very few observations.

10. *The dispersal of the differences is considerably higher for IGRA matchups, partly due to longer time differences allowed, and the drift of the sondes during the ascent.* Higher differences may due to the radiosonde ascents drift and vertical extent will be different over different geographical domains. Similarly, the collocations matchups of clear sky pixel retrievals will vary and hence the differences values also vary latitudinal.

Thank you for your suggested explanation. Discussion of geographical variance of sonde drift was added to section 3.1 (lines 104-108) and to end of section 5.1.

*Observed in SUOMINET comparisons, the bias reduction can be related to radiosonde data or to collocation criteria. General comparisons also indicate very good agreement between OLCI-A and -B.*
What is retrieval algorithm of IWV of OLCI-A and -B. How bias can be reduce and related to radiosonde data.

Here we discuss the reasons for lower bias observed in radiosonde comparisons at very high IWV values. We clarified the text to avoid the confusion. New text reads "*the dispersal of the*

*differences is considerably higher for IGRA matchups, partly due to longer time differences allowed, and the drift of the sondes during the ascent*" (Lines 163-165).

Line120: *INLAND_WATER water pixels, representing rivers and lakes, similarly show wet bias and large dispersal.*
Give references.

The INLAND_WATER pixels were analyzed separately as part of this study, but the scatterplot is not shown here. The dispersal can be seen in figure 7 (of the initial submission). Following the recommendation of reviewer 2, the figure and the discussion of water pixels was removed from the revised manuscript.

*The dependency observed for latitude and solar zenith angle is related to generally higher water vapour total columns seen in low latitudes and solar zenith angles,while the seasonal cycle is consistent with the over-presentation of northern hemisphere stations and higher total columns during summer months.*

Give some prove or reference for this claim?

These are our interpretations from the data shown here. We stress this in the revised version. Timeseries and the discussion of the seasonal cycle was removed from the revised version as per recommendation by reviewer 2.

Line 161-168: *OLCI observations classified as water surfaces (WATER and INLAND flags, including TIDAL with WATER) considerably larger bias and dispersal than those classified as land surfaces (LAND flag, including TIDAL with LAND).*

OLCI observations in pixels data contains both sea and mountainous land together along with topographically diverse terrains around these stations may introduce large bias.

That is true. Unfortunately for this study, the meteorological stations are often located in "interesting" locations, presenting a problem for the representativeness of the station. Small footprint of the modern satellite instrument like OLCI can partially offset this issue. WATER pixels are not considered in the revised version of the manuscript, further mitigating this issue.

We added a note in the revised version.

Technical Corrections:

Line 90: May need prove reading carefully.

Section was rewritten to improve clarity.

---

## Author Comment (AC2)

Kalakoski et. al.: Validation of Copernicus Sentinel-3/OLCI Level 2 LAND Integrated Water Vapour product

The paper presents the analysis of validation of the Integrated Water Vapour (IWV) from Sentinel-3 Ocean and Land Colour Instrument (OLCI). The statistics is based on the comparison with data from the SUOMINET network and from the IGRA database. In my opinion, the scientific value of this research is rather high since IWV is the atmospheric parameter required for solving a large number of problems of climate change and weather prediction. Besides, the scientific interest to the described results is determined by the fact that the presented analysis is comparative (the data are provided by three independent sources) and covers different geographical locations. The obtained results could be helpful for the improvement of the retrieval algorithm used to process the OLCI measurements.

Despite the fact that the paper contains valuable and interesting results, I can not recommend it for publication in the present form. **The paper requires major revision.** There are several issues of general character which should be addressed in this paper. Besides, there are also specific remarks.

Authors thank the reviewer for careful reading and constructive comments. The suggestions given, especially for the results section, help to make the paper more coherent and focused.

Please find below our answers (in blue) to the comments presented (in black).

**General critical issues:**

1) The IWV measurements by OLCI are not placed within the context of modern space-based observations of IWV. Such placing is especially important since providing information on atmospheric parameters is a secondary goal of the OLCI mission and not a primary goal as the authors indicate in the Introduction section. My recommendation is to give a comparison of the IWV retrieval accuracy values and ground pixel size values which are declared or estimated for OLCI and other satellite instruments which measure IWV. Such comparison would help to assess the value of the IWV observations by OLCI.

Section on other satellite sources with comparison to OLCI was added to the introduction (Lines 22-42).

2) Section 2 "OLCI Integrated Water Vapour retrieval" is too sketchy. I recommend giving more details of the retrieval algorithm instead of giving only references to technical guidelines. In addition, full resolution and reduced resolution modes should be explained which are now only briefly mentioned (Page 2, line 46).

Section describing the algorithm and retrieval was extended.

3) The description of the data selection procedure in Section 4 is very unclear. In particular, the sentence "Missing satellite extractions..." (Page 4, Line 88) can be misleading. Please explain what you mean when you say that the reference observations should be validated (Page 4, Lines 90-91). I recommend the authors to remove current subsections of Section 4 and to provide the information on data selection and quality control on step-by-step basis

separately for IGRA and SUOMINET. All criteria for match-up should be given clearly. The reasons for choosing 31x31 macropixels should be presented also.

Agreed, the data selection section is unclear. The section 4 was written in subsections to separate selection done at the database creation (4.1) and selection done during the analysis (4.2). As it is, this distinction is not clear.

Line 88: Agreed, clarified in the revised version.

Lines 90-91: Validated here is only taken to mean "fulfill the quality criteria".

Section 4 was reorganized and clarified in the revised manuscript as suggested.

4) In my opinion, the presentation of the results in Section 5 requires improvement:

- The cloud-screening criteria (Page 4, Line 102) have been already described in Section 4. Mentioning these criteria in Section 5.1 can produce a wrong impression that they were used not in all cases.

Agreed, cloud-screening criteria are repeated unnecessarily. Repeated discussion is removed from the revised version of section 5.

In order to avoid confusion, I recommend not to mention the results with WATER pixels. The authors note that they make validation for land pixels and "WATER pixels are not strictly part of the ESA OLCI product".

Agreed, discussion of WATER pixels was removed from the revised version of the manuscript.

I can not understand whether figures A1 and A2 belong to any Attachment or not. I would like to see these figures and relevant expanded discussion within the main text. I am not sure if bottom rows in these figures are necessary since all latitudes are taken together and therefore seasonal dependence seems to be hard to detect.

We agree with the reviewer that the status of figure A1 and A2 is unclear. Figures are moved to Section 5 in the revised version (figures 4 and 5) with revised discussion (lines 175-179). Likewise, we agree that the seasonal dependency (or lack of it) is difficult to see in the bottom panels. Accordingly, the bottom panels are removed from the revised versions of the figures.

Besides, I have the feeling that Fig. 3 duplicates information which is already contained in Fig. 2.

Information on Figure 3 can indeed also be seen in Figure 2. However, we feel that the presentation in Figure 3 is clearer to some readers.

- The title of Subsection 5.2 "Classification of biases" is misleading because mainly the problem with camera 3 is discussed. I am not sure if Fig. 5 is necessary. It is sufficient to mention in the main text about negligible influence of neighbouring cloud-flagged pixels.

Agreed, we changed the title of the subsection and removed the figure 5. Mention of the cloud proximity was added to section 5.1 (lines 172-174) with the figure 5 provided as supplementary material.

- In Subsection 5.3 please explain how the distribution of the error estimates from the retrieval algorithm was obtained. I suppose that every single OLCI measurement has its own error estimation and you just collected this information from all OLCI measurements. Is it so?

Yes, each OLCI measurement has an associated uncertainty estimate. The bottom-left panels of Figure 6 show the mean value of the error estimate for pixels within the macro-pixel.

- I would like to recommend removing Subsection 5.4 and Fig. 7 in order to avoid confusion. As I have already mentioned above, the reason is that water pixels are not included in the OLCI products. If the authors decide to keep this subsection, all terms should be explained (WATER, TIDAL, INLAND), the number of observations in this critical pixels should be given, the observational conditions should be described (water, ice, ice covered with snow) etc.

Agreed, the emphasis of the paper is on the LAND observations. The discussion of WATER pixels and Figure 7 was removed from the revised manuscript.

**Specific remarks:**

Page 1, Line 2
What is the meaning of the term "geophysical" in the context of validation?

The word "geophysical" is removed to avoid confusion.

Page 1, Line 15
Remove the abbreviation ECV which is not used in the text below.

Removed accordingly

Section 3.1
Is it possible to roughly estimate the typical horizontal drift of a radiosonde during ascent to the pressure level of 500 hPa in order to assess the "effective" horizontal resolution of radiosonde measurements of IWV?

Median horizontal drift of the soundings is about 10 km, depending on the latitude (Seidel et. al., 2011). We added this information in the revised version (lines 104-108).

Section 3.2
If possible, please provide the information about pressure level which is assumed as an upper limit for IWV in the SUOMINET GNSS network? What is the area of horizontal averaging for the derivation of IWV by the GNSS method?

GNSS IWV is assumed here to represent full atmospheric column. If any upper limit exists, we assume it to be high enough to have little or no systematic effect on the GNSS derived IWV.

The horizontal resolution of the GNSS method depends on the azimuth angle of the GNSS satellites visible at the time of the observation. In general, Van Malderen et. al., (2014) consider the GNSS observation to represent a cone with area of roughly 100 km$^2$. Value and the reference were added to Section 3.2 (lines 116-117).

Page 4, Line 82
Please explain the acronym ACRI-ST.

To our knowledge, ACRI-ST is not an acronym.

Page 5, Line 112
Please compare typical time of the sonde ascents to the level of 500 hPA with the allowed time mismatch between OLCI and sonde observations.

Median descent time to 500 hPa is 30 minutes (Seidel et al. 2011). Discussion is added to the revised version of Section 3.2 (lines 107-108).

REFERENCES

Seidel, D. J., Sun, B., Pettey, M., & Reale, A. (2011). Global radiosonde balloon drift statistics. *Journal of Geophysical Research: Atmospheres*, *116*(D7).

Van Malderen, R., Brenot, H., Pottiaux, E., Beirle, S., Hermans, C., De Mazière, M., ... & Bruyninx, C. (2014). A multi-site intercomparison of integrated water vapour observations for climate change analysis. *Atmospheric Measurement Techniques*, *7*(8), 2487-2512.

---

## Referee Report (RR1)

**Comments on the revised version of "Validation of Copernicus Sentinel-3/OLCI Level 2 LAND Integrated Water Vapour product" by Niilo Kalakoski et al.**

**Anonymous reviewer**
**8 July 2022**

The authors have changed the manuscript taking into account all comments and suggestions. In the revised version, the presentation of the input data and the results became clear and unambiguous. Still, I have few minor comments which I recommend to consider before submitting the final version for publication:

Page 5, Line 136
From this sentence in Section 4.1, I assume that initially all matchups in the database fit the +/- 3 hour time window, both for IGRA and SUOMINET. In Section 4.2 the time window for SUOMINET matchups was reduced to 15 minutes. In order to clarify this issue I would recommend making an explicit corresponding note in the text of Section 4.1.

Captions for Figure 4 and Figure 5
In my opinion, these captions can be misleading. The authors write:
*"Solid line shows the median of each 2 kg/m² wide bin,……. Bins with less than 20 matchups were omitted from the figure"*
I guess that the latitude bins and the SZA (solar zenith angle) bins are used in these figures, but not the IWV bins. If so, please remove mentioning the IWV bins and indicate the width of the latitude and SZA bins.

In the revised version, the authors analyse the IWV results only over land. Though this fact is indicated in the title and in the abstract of the manuscript, I would recommend emphasizing it somewhere in the main text also.

Page 1, Line 17
The authors agreed to remove the abbreviation ECV which is used only once but it is still present in the text.